# Simple discrete-time self-exciting models can describe complex dynamic processes: A case study of COVID-19

Raiha Browning[1,2]*, Deborah Sulem[3], Kerrie Mengersen[1,2], Vincent Rivoirard[4], Judith Rousseau[3,4]

**1** School of Mathematical Sciences, Queensland University of Technology, Brisbane, Australia, **2** Australian Research Council, Centre of Excellence for Mathematical and Statistical Frontiers, Brisbane, Australia, **3** Department of Statistics, University of Oxford, Oxford, United Kingdom, **4** Ceremade, Université Paris-Dauphine, Paris, France

☉ These authors contributed equally to this work.
* raihatuitaura.browning@qut.edu.au

**Data Availability Statement:** The data used in this analysis are available on Github: https://github.com/RaihaTuiTaura/covid-hawkes-paper. This data was obtained from Johns Hopkins University:

## Abstract

Hawkes processes are a form of self-exciting process that has been used in numerous applications, including neuroscience, seismology, and terrorism. While these self-exciting processes have a simple formulation, they can model incredibly complex phenomena. Traditionally Hawkes processes are a continuous-time process, however we enable these models to be applied to a wider range of problems by considering a discrete-time variant of Hawkes processes. We illustrate this through the novel coronavirus disease (COVID-19) as a substantive case study. While alternative models, such as compartmental and growth curve models, have been widely applied to the COVID-19 epidemic, the use of discrete-time Hawkes processes allows us to gain alternative insights. This paper evaluates the capability of discrete-time Hawkes processes by modelling daily mortality counts as distinct phases in the COVID-19 outbreak. We first consider the initial stage of exponential growth and the subsequent decline as preventative measures become effective. We then explore subsequent phases with more recent data. Various countries that have been adversely affected by the epidemic are considered, namely, Brazil, China, France, Germany, India, Italy, Spain, Sweden, the United Kingdom and the United States. These countries are all unique concerning the spread of the virus and their corresponding response measures. However, we find that this simple model is useful in accurately capturing the dynamics of the process, despite hidden interactions that are not directly modelled due to their complexity, and differences both within and between countries. The utility of this model is not confined to the current COVID-19 epidemic, rather this model could explain many other complex phenomena. It is of interest to have simple models that adequately describe these complex processes with unknown dynamics. As models become more complex, a simpler representation of the process can be desirable for the sake of parsimony.

https://github.com/CSSEGISandData/COVID-19/
tree/master/csse_covid_19_data/csse_covid_19_
time_series.

**Funding:** The author(s) received no specific
funding for this work.

**Competing interests:** The authors have declared
that no competing interests exist.

## Introduction

The outbreak of the novel 2019 coronavirus disease (COVID-19) was declared a Global Health Emergency of International Concern on 30th January 2020, and pronounced a Pandemic on 11th March 2020. It has since spread rapidly with over 116 million confirmed cases and more than 2.5 million deaths as of 7th March 2021 [1]. Since the first reported case in December 2019, countries around the world have fought to contain the virus. In the absence of a vaccine, countries implemented a range of non-pharmaceutical interventions and strategies to reduce the spread of the virus, from measures such as social distancing, mask-wearing and contact tracing, to complete city lockdowns and stay at home orders. These recommendations are guided by mathematical and statistical modelling to quantify the efficacy of these measures [2–9].

There is now an expansive collection of research dedicated to understanding the virus from all perspectives, including its biological, epidemiological, clinical, economic and social impacts. There is also a wealth of knowledge around prevention strategies to control the outbreak. In all of these, statistical and mathematical models are an essential aspect to gaining meaningful insights into how the virus spreads and quantifying its various impacts. A popular choice is compartmental models, with some considering the standard SIR (Susceptible-Infected-Recovered) model [10–12], and further extensions in which additional states are introduced [13–18]. As an alternative to compartmental models, others have used methods such as branching processes to capture the spread of the virus through individual networks [2, 3, 5], log-linear Poisson autoregressive models [19], and other probabilistic models of the infection cycle of the virus [20]. Various models based on growth curves have also been proposed, for example [21–23], who use logistic, exponential and Richards growth curves respectively. More detailed approaches such as agent-based modelling have also been considered by numerous authors [24–27].

A Hawkes process [28] is a stochastic, self-exciting process in which past events influence the short-term probability of future events occurring. They are often used to explain many phenomena that exhibit self-exciting properties, including neuroscience [29–31], crime and terrorism [32–34], seismic activity [35] and social media [36]. Similarly, due to their contagious nature it is also natural to represent infectious diseases, such as the current COVID-19 pandemic, as a Hawkes process.

Hawkes processes have been successfully applied to model epidemics and infectious diseases. For example, for the Ebola outbreaks in West Africa and the Democratic Republic of Congo [37, 38], the Hawkes process is found to outperform the SEIR (Susceptible-Exposed-Infected-Recovered) mechanistic model in terms of short term prediction. Another study employs an extension of the multivariate Hawkes process to understand the transmission routes and regional connectivity for the dengue fever outbreak across regions in Australia [39]. Rocky Mountain Spotty Fever has also been modelled using a recursive Hawkes process, with the expected number of transmissions based on the current conditional intensity of the Hawkes process [40]. Moreover [41], model invasive meningococcal disease using a spatiotemporal extension to the Hawkes process.

The spread of COVID-19 is an extremely complex process, with unknown disease dynamics and huge variations in the preventative measures and responses of different countries. We propose a parsimonious model for COVID-19 deaths, namely discrete-time Hawkes processes (DTHP) [32, 33, 42], to describe the complicated dynamics of the COVID-19 epidemic. In its original form, the Hawkes process is a continuous-time point process; however, the DTHP observes the occurrence of events at a discrete time resolution. Due to this construction, the DTHP can directly model the available data (i.e. daily counts), without artificially imputing the

data onto a continuous timeline, as is generally done in studies using continuous-time Hawkes processes. We also introduce deterministic change points in this study, since the dynamics of the spread vary abruptly as the pandemic progresses and preventative interventions are introduced.

Alternative models, such as the mechanistic and growth curve models discussed previously, primarily focus on estimating the model parameters that govern the system. Hawkes processes, however, are more detailed, as individual events and their respective occurrence times directly influence the likelihood of future events occurring. Hawkes processes also provide additional insights into the infection dynamics of diseases by estimating the level of external cases through the baseline parameter and the triggering kernel, which models the decay in infectivity through time.

Hawkes processes and compartmental models are based on different mathematical principles and rely on different assumptions. However, their connection was explored by [43]. These authors show that, via a modified, finite population variant of the Hawkes model for a particular choice of triggering kernel, the rate of events is equivalent to the SIR model's infection rate. While the SIR family of models is useful if more is known about the system dynamics, a simpler model is often useful for phenomena where there are many unknowns. We show in this study that our model is helpful for this purpose. Additionally, we explore the differences between Hawkes, compartmental models and other approaches further in the discussion.

## Related work

An approach to modelling the COVID-19 pandemic using self-exciting branching processes has been suggested by [44]. These authors employ a continuous-time Hawkes model with a nonparametric estimate of the reproduction number, $R(t)$, the average number of secondary cases produced by a single case of the virus. Both death counts and the number of confirmed cases in the early stage of the epidemic, before April 1st, are modelled in three states of the U.S., several European countries and China. Compared to SIR and SEIR models with a fixed reproduction number, their Hawkes model with a dynamic parameter leads to lower estimates of the basic reproduction number, $R_0$. In the same line of work [45], consider several datasets for the state of Indiana in the early stage of the epidemic. They also compare a nonparametric estimate of the reproduction number, $R(t)$, with an exponentially decreasing function and a step-function, and find that the estimation of $R$ is very sensitive to the type of input data (i.e. deaths or cases), the data source, and the model choice. Similarly [46], adopt a continuous-time Hawkes model with spatial covariates to model both the number of confirmed COVID-19 cases and the number of deaths, for the U.S. at the county level. This study also considers a time-varying reproduction number. Finally [47], also use the continuous-time Hawkes process to illustrate the severity of the virus in France if no preventative action were to be taken.

Two similar approaches to ours are that of [48, 49]. The former proposes a two-phase contagion model based on an extension of the Hawkes process. This study considers a continuous-time Hawkes process, assume the rate of external events varies through time, and estimate the change point in their model. The authors also assume there is no external excitation after the change point. The latter of these is, to the authors' knowledge, the most similar approach to ours. These authors consider a discrete-time Hawkes process to describe the current COVID-19 epidemic. This study focusses on estimating a time-varying reproduction number, ignoring the influence of external activity and considering a fixed excitation kernel.

Several other approaches for modelling COVID-19 that incorporate change points have been proposed to capture the dynamic nature of the pandemic. [50, 51] find that using compartmental models with time-varying infection rates, the estimated change points for Germany

and South Africa, respectively, align with various government interventions in these countries. [52] do not directly estimate the change points; instead, they propose a compartmental model for Italy with piecewise model parameters partitioned into regular time intervals. Alternatively [53], consider a combination of exponential and polynomial regression models to estimate the optimal change points for the COVID-19 outbreak in India. While these studies consider only a single country [54], examine several countries and introduce a single stochastic change point into their compartmental model. [55] present a widespread study across 55 countries using a partially observed Markov process with piecewise transmission rates.

## Contributions

In the current literature, the continuous-time Hawkes process requires artificial imputation of the daily count data onto a continuous time resolution, adding a significant computational burden to the implementation and adding additional, potentially unnecessary, noise to the model. We develop a multi-phase approach for the DTHP to directly model the reported daily counts of the number of deaths caused by the virus.

The dynamics of the process before and after the enactment of preventative measures and policy interventions to reduce the spread of the virus are inherently different. The majority of the existing literature on modelling the COVID-19 pandemic using Hawkes processes consider only the early stages of the pandemic. In this work, we develop a variant of the DTHP to model the distinct phases of the COVID-19 epidemic. We modify the traditional Hawkes process to account for this change in dynamics by including deterministic change points in the model.

While [49] also study more recent data, these authors limit parameter estimation to the reproduction number, and fix the remaining parameters of the Hawkes model. In our study, we estimate the excitation kernel for additional flexibility. Regarding external events [48], also assume there is no external excitation in the second phase of their two-phase model. We make no such assumption, and believe considering external excitation throughout the entire course of the pandemic is a valuable consideration. There are still travellers arriving from abroad, and thus exogenous activity is still occurring in later phases at a lower rate. This is particularly relevant as many countries have relatively relaxed quarantine requirements, which means that travellers from abroad are still capable of spreading the virus. Although we study mortality data in this analysis, we are able to make a connection between mortalities and infections. In particular, we show in S1 Appendix that the rate of external events in our model can roughly be interpreted as external infections, times the probability of death given infection. This link is particularly useful in the absence of reliable infection data.

Change point models for Hawkes processes have been considered in other applications [56]. However, these authors assume independence of the observed data between change points, prohibiting events that occur within a time period to influence events in future time periods. This type of model is inappropriate for this application, as the time periods are not independent. While the behaviour of the process varies between time periods, the influence of past events remains active in the memory of the process. Thus, the baseline parameters become artificially inflated if events from different time periods are assumed to be independent. For the current COVID-19 pandemic [49], introduce a method for detecting change points in the reproduction number through augmenting their Hawkes model with state-space methods.

In particular for the COVID-19 epidemic, while other studies directly estimate the change points or partition the timeline into regular intervals to reflect the evolving dynamics of the epidemic, we propose a simple method that incorporates fixed change points. We do not estimate the change points for our model, as it was fairly obvious where a reasonable change point

was in these data, and this avoids complexity arising from different interventions being introduced in each country, with varying levels of restrictions. Furthermore, the delays before tangible results are observed, in addition to the complex and hidden interactions underlying the process, complicate the interpretation of estimated change points. We instead opt for this consistent and simplistic definition of the change point for each country. The change points could however be estimated for more complex trajectories.

We illustrate in this study how a simple model can be used to describe exceedingly complex natural phenomena such as epidemics, and in particular the COVID-19 pandemic. Although it is the same underlying phenomenon, all countries are unique concerning the spread of the virus and the resultant response measures. Our simple model can capture these dynamics. Additionally, while many other studies consider small-scale regions, such as individual counties in the U.S., we are also able to gain insights into the dynamics of the process at a higher-level across entire countries.

### Outline

First we define a general form of the DTHP, and contrast this with its continuous-time equivalent. We then introduce the particular model used in the initial stage of this analysis for modelling COVID-19, incorporating a change point into the construction of the DTHP. Next, a brief description of the data and inference methods are provided. Finally, the results for the ten countries of interest are presented, and we also show the results from fitting our model to more recent data. This is followed by a discussion and concluding remarks.

## Methods

### Discrete-time Hawkes process

The discrete-time Hawkes process is a self-exciting stochastic process whereby events occur at regular intervals on a discrete-time scale. It follows a similar construction to the continuous-time Hawkes process [28]. The conditional intensity function $\lambda(t)$ characterises a Hawkes process, and herein lies the difference between the continuous-time and discrete-time variants. For the DTHP, $\lambda(t)$ represents the expected number of events that occur at time interval $t$, conditionally on the past. In contrast, for the continuous-time Hawkes process, $\lambda(t)$ is the instantaneous rate of an event occurring at time $t$. The DTHP model also has an extra layer of flexibility compared to its continuous-time counterpart as the underlying data generating process can be selected as any counting distribution with conditional mean $\lambda(t)$.

Consider a linear univariate discrete-time Hawkes process $N$, where $N(t)$ represents the number of events up to time interval $t$. $N(t)$ is dependent on the history of events up to but not including time $t$, denoted by $H_{t-1} = \{y_s : s \leq t - 1\}$, where $y_s$ represents the observed number of events in a given time interval $s$. Furthermore, $N(t) - N(t - 1)$ represents the number of event occurrences at time $t$, and thus,

$$\begin{aligned} \lambda(t) &= E\{N(t) - N(t-1)|H_{t-1}\} \\ &= \mu + \alpha \sum_{i:t_i < t} y_{t_i} g(t - t_i) \end{aligned} \tag{1}$$

where $\mu$ represents the baseline mean of the process and the second term represents the self-exciting component of the Hawkes process, describing the expected number of events during a particular interval $t$ given previous events. The triggering kernel $g(t - t_i)$ describes the influence of past events on the intensity of the process, given the time elapsed since event $i$, where $t > t_i$. In this study, we specify the triggering kernel to be a proper probability mass function

with strictly positive integer-valued support. Since the sum of the excitation kernel over $\mathbb{Z}_+$ is equal to 1, one can interpret the non-negative magnitude parameter $\alpha \in \mathbb{R}_{\geq 0}$ as the expected number of subsequent events produced by a single event [33].

## Model

Daily counts of the reported number of deaths of the novel coronavirus COVID-19 are modelled using the discrete-time Hawkes process, where the number of events observed on day $t$, namely $y_t$, are distributed according to the random variable, $Y(t)$, which has conditional mean $E(Y(t)|H_{t-1}) = \lambda(t)$ as defined in Eq (1). In this analysis $Y(t)$ is assumed Poisson distributed, thus $Y(t) \sim \mathcal{P}(\lambda(t))$. The Poisson distribution is selected as it has an intuitive interpretation regarding the generation of daily death counts on a given day, and because it is a natural approximation of a binomial distribution with a large population and low death rate. More detail is given in S1 Appendix. Thus, for the proposed DTHP model, the probability that day $t$ has $y$ events is,

$$P(Y(t) = y|\lambda(t)) = \frac{\lambda(t)^y e^{-\lambda(t)}}{y!}$$

First we consider an initial period up to 25th July 2020, to determine some initial modelling assumptions and study the model performance in the early stages of the pandemic. The conditional intensity function $\lambda(t)$ is altered from Eq (1) to allow for a change point in the process, since the DTHP with fixed parameters is unable to capture the complex dynamics for an epidemic of this scale. The parameters of the DTHP implicitly incorporate environmental and social characteristics that are significant for the spread of the disease, and these characteristics change after preventative measures are introduced. Thus, if the dynamic nature of the epidemic is not taken into account, the model averages the estimated parameters, combining the effects of the initial explosive phase of the pandemic with the downward trend that follows after the implementation of preventative measures.

In the initial period of analysis, to accommodate this shape, we assume in our analysis that two phases can adequately separate the underlying dynamics. Namely, these phases are the initial period where the virus is spreading rapidly and the following period of reduced contagion resulting from the introduction of preventative measures and policies. Many complex interactions are occurring in the deaths process. For example, as medical professionals become more familiar with the virus and treatments are improved, medical facilities are better equipped to deal with COVID-19 patients in critical condition requiring ICU [57, 58]. However, this can be offset by increased demand for hospital beds, resulting in medical facilities becoming overwhelmed and unable to care for all patients that require hospital treatment. Therefore, rather than making explicit assumptions about the underlying processes driving the death dynamics, we link our Hawkes model on the death dynamics to a similar infection model, as we discuss in S1 Appendix.

Thus, we first retrospectively define a single change point at time $T_1$, where $T_1$ is the maximum value of deaths, to capture the different dynamics of the epidemic at two distinct stages of the outbreak.

The triggering kernel $g(t - t_i)$ is selected as a geometric excitation kernel, $g(t - t_i; \beta) = \beta(1 - \beta)^{t - t_i - 1}$. The exponential distribution is one of the most commonly used triggering kernels for continuous-time processes. Thus we choose the geometric kernel as it can be shown to be equivalent to the exponential distribution in the context of discrete time. The parameter $\beta$ represents the success probability in the geometric distribution, and thus the average of the

excitation kernel is $\frac{1}{\beta}$. We also express the expectation of the maximum excitation time in terms of the parameters of the model in S2 Appendix.

The conditional intensity function before $T_1$ is calculated using one set of model parameters, $(\mu_1, \alpha_1, \beta_1)$. After $T_1$, the intensity function is calculated using a new set of parameters, $(\mu_2, \alpha_2, \beta_2)$ for the second phase in the epidemic. Thus for one change point at time $T_1$, $\lambda(t)$ is given by,

$$\lambda(t) = \begin{cases} \mu_1 + \alpha_1 \sum_{i:t_i<t} y_{t_i} g_1(t - t_i), & t \leq T_1 \\ \mu_2 + \alpha_2 \sum_{i:t_i<t} y_{t_i} g_2(t - t_i), & t > T_1 \end{cases} \tag{2}$$

It is straightforward to extend Eq (2) to allow for additional change points. While the majority of this paper considers only the initial stage of the pandemic up to 25th July 2020, we consider subsequent phases after this date as a set of additional analysis. This is to demonstrate how our model can be extended beyond the initial phases of the pandemic, as new data will continue to become available each day for the foreseeable future.

Although we consider the deceased population rather than the infected population, there is a connection between the two under some simplifications. Thus studying deaths is useful for understanding the infection dynamics as well. This is advantageous particularly in the early stages of a pandemic, when no reliable data on infections are available. We do not go into the details here, but the key outcome of this is that $\alpha$, $\beta$ and a function of $\mu$ are interpreted with respect to infections, not deaths. The full derivation is available in S1 Appendix. As this approximation relies on the assumption of a large population and a low death rate, we would not expect this model to be reasonable for other time series where the rate of occurrence is high, such as COVID-19 recoveries.

For a time series of $T$ days and a given country, the log-likelihood function for this DTHP model with retrospective change point, $T_1$, up to an additive constant $K$, is then,

$$\log L(\boldsymbol{y}|\boldsymbol{\mu}, \boldsymbol{\alpha}, \boldsymbol{\beta}) =$$

$$K + \sum_{t=1}^{T_1} \left[ y_t \log \left( \mu_1 + \alpha_1 \sum_{i:t_i<t} y_{t_i} \beta_1 (1 - \beta_1)^{t-t_i-1} \right) - \left( \mu_1 + \alpha_1 \sum_{i:t_i<t} y_{t_i} \beta_1 (1 - \beta_1)^{t-t_i-1} \right) \right]$$

$$+ \sum_{t=T_1+1}^{T} \left[ y_t \log \left( \mu_2 + \alpha_2 \sum_{i:t_i<t} y_{t_i} \beta_2 (1 - \beta_2)^{t-t_i-1} \right) - \left( \mu_2 + \alpha_2 \sum_{i:t_i<t} y_{t_i} \beta_2 (1 - \beta_2)^{t-t_i-1} \right) \right]$$

## Data

We use data gathered by the Johns Hopkins University [59] in this work. These data come in the form of daily counts of confirmed cases or deaths by country and region. In this analysis, the number of daily reported deaths for a selection of countries, namely Brazil, China, France, Germany, India, Italy, Spain, Sweden, the United Kingdom and the United States, are considered. We select these countries to represent a global sample of countries that have been adversely affected by the coronavirus outbreak. It is important to note that the definition of deaths due to COVID-19 varies between countries. These differences are ignored in our modelling.

The reported number of deaths was considered a more reliable response variable than the reported number of cases. This is due to data issues that can arise when considering the number of confirmed cases, such as lack of testing or differing testing rates between countries, differences in definitions and differences in the timing for reporting of cases. Additionally, to

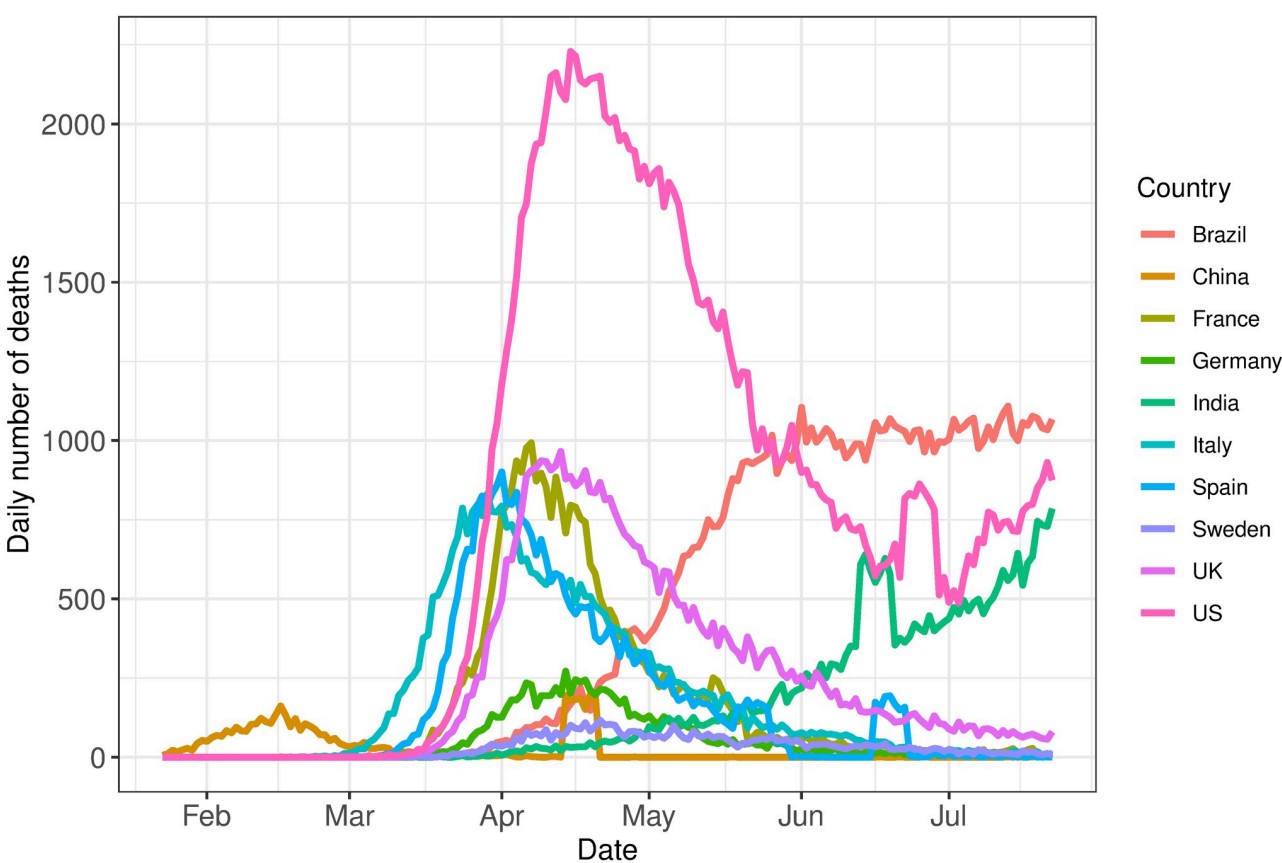

**Fig 1. Observed data.** Daily volume of deaths due to COVID-19 for the countries selected in this analysis.

mitigate the effect of systematic influences in reporting, such as lower reporting on weekends [50], the data is smoothed over a rolling window of seven days. The start of the observation window, $t_1$, for each country is defined as the time the number of deaths exceeds ten. Fig 1 shows the smoothed volume of daily deaths for the countries under consideration up to 25th July 2020.

For the initial stage of this analysis, we consider data up to 25th July 2020. We define a single change point, $T_1$, as the time where the maximum number of deaths occurs, for the countries with sufficient data in the downward phase of the epidemic by the end of the initial study period. Where there is insufficient evidence for the downward trend, for example, in India and Brazil, no change point was introduced, and only a single phase was modelled. Moreover, the trend for Brazil showed evidence of the curve flattening; however, there was insufficient data for this second phase. Thus the end of the observation window for Brazil is fixed on 1st June 2020. Additionally, as China, India, Spain and the United States experienced large deviations from the current trend towards the end of the observed data, earlier endpoints of 13th April 2020, 12th June 2020, 15th June 2020 and 21st June 2020 were imposed respectively. This avoids the anomalous spikes at the end of these series, since it was not clear whether these aberrations were real or due to reporting definitions or other errors. The endpoint for the remaining countries was set as 25th July 2020. We later extend our analysis to include more recent data, to demonstrate the utility of our model in later phases of the pandemic. A description of the data processing for this is in the relevant Results section.

## Parameter inference

Parameter estimation is undertaken using Bayesian methods. We consider a range of prior choices for the baseline parameters $\mu_1$ and $\mu_2$, and perform leave-future-out cross validation with Pareto smoothed importance sampling [60] to assess the performance of each prior choice. The priors considered are,

$$
\mu_1, \mu_2 \quad \sim \quad
\begin{cases}
\log N(1, 1) \\[4pt]
\log N(5, 1.5) \\[4pt]
\mathrm{Gamma}(2, 2) \\[4pt]
\mathrm{Gamma}(5, 1) \\[4pt]
U(0, \infty),
\end{cases}
$$

where the first term of the log-normal priors represents the mean of the random variable itself, as opposed to the mean of the variable's natural logarithm.

Cross validation with Pareto smoothed importance sampling relies on the expected log predictive density (ELPD), for which a larger value indicates a better model fit. We calculate the ELPD in each country for each of the baseline parameter prior choices, and these results are provided in S1 Table. Based on this analysis, there is no obvious choice of prior that consistently outperforms the rest for each country. On the contrary, the difference in the ELPD is marginal between priors. The remainder of this paper presents the results for $\mu_1, \mu_2 \sim \mathrm{Gamma}(5, 1)$, as this is most frequently the highest ELPD, and if not the maximum, is generally very comparable.

Flat priors are selected for $\alpha_1, \alpha_2, \beta_1$ and $\beta_2$ such that,

- $\pi(\alpha_1, \alpha_2) \propto \mathbb{I}_{(0,\infty)^2}(\alpha_1, \alpha_2)$

- $\beta_1, \beta_2 \sim U(0, 1)$

A Metropolis-adjusted Langevin step [61] is used to jointly update $\alpha_1$ and $\beta_1$, and also to jointly update $\alpha_2$ and $\beta_2$. Denoting the parameters at iteration $t$ by $\alpha^{(t)}, \beta^{(t)}$, the proposals $\alpha^*, \beta^*$ are simulated from,

$$
\begin{bmatrix} \alpha^* \\ \beta^* \end{bmatrix} \sim \mathcal{N}\left( \begin{bmatrix} \alpha^{(t)} \\ \beta^{(t)} \end{bmatrix} + \frac{\epsilon^2}{2} G \begin{bmatrix} D_\alpha(\alpha^{(t)}, \beta^{(t)}) \\ D_\beta(\alpha^{(t)}, \beta^{(t)}) \end{bmatrix}, \epsilon^2 G \right)
\tag{3}
$$

where $D_\alpha(.)$ and $D_\beta(.)$ are the gradients of $\log L$ with respect to $\alpha$ and $\beta$ respectively, $G$ is a pre-conditioning matrix accounting for covariance between parameters and $\epsilon$ is the step size in the Metropolis-adjusted Langevin algorithm.

The MCMC chain was run for 60,000 iterations discarding the first 20,000. The pre-conditioning matrix $G$ was taken as the covariance matrix from an implementation of the standard Metropolis-Hastings algorithm for each country. The R code and data required to replicate this study are available on Github (https://github.com/RaihaTuiTaura/covid-hawkes-paper).

## Results

We first present results from the initial analysis considering data up to 25th July 2020. Fig 2 presents the 95% posterior intervals around the estimated conditional intensity function $\lambda(t)$ against the observed data for each country. The estimated intensity function on day $t$,

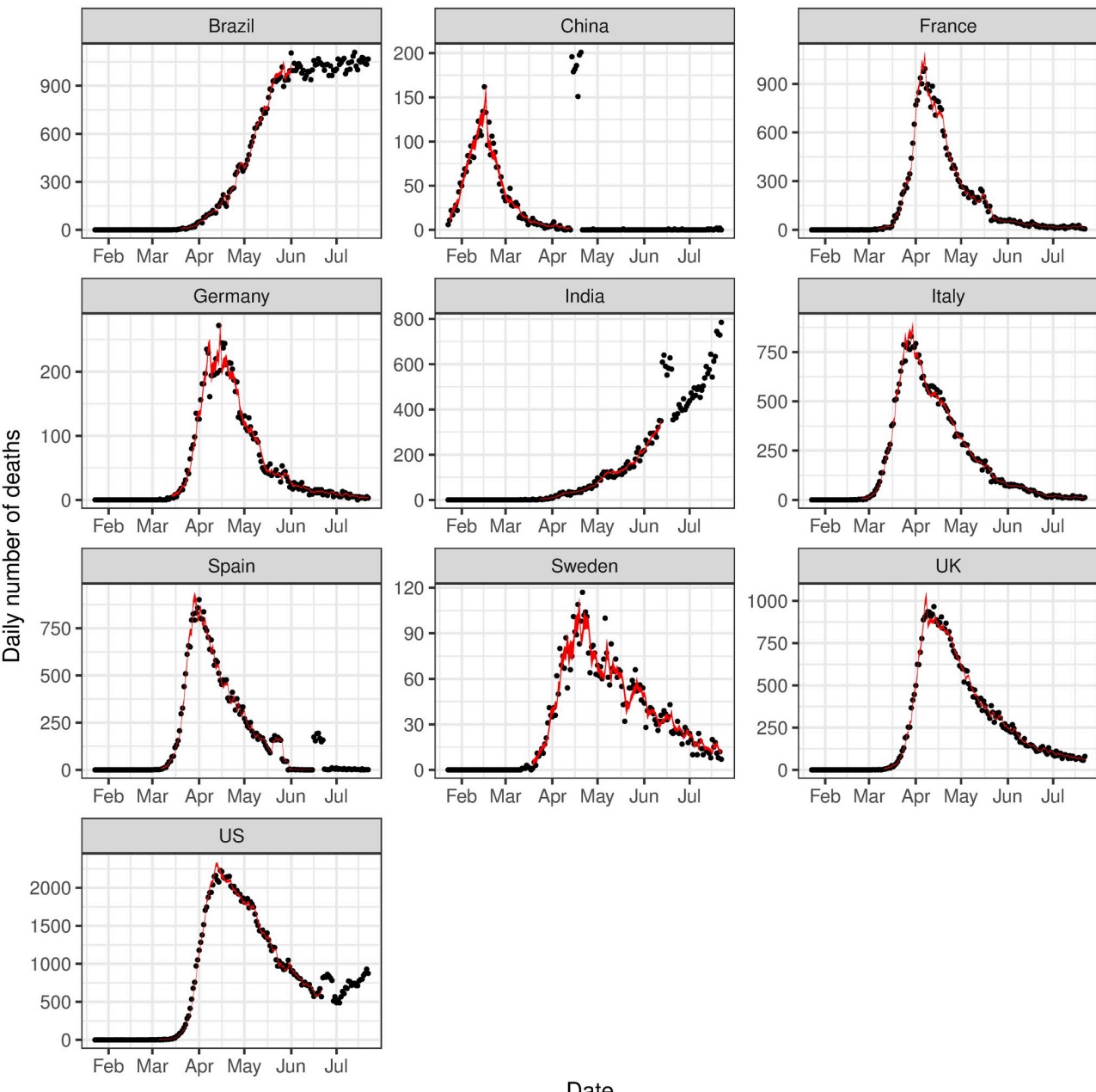

**Fig 2. Observed deaths versus estimated deaths.** The observed number of deaths (black dots) compared to the 95% posterior interval for the estimated expected number of events, i.e. λ(*t*) (solid red ribbon).

represents the expected number of events on day *t* and very closely follows the observed number of deaths. It is also extremely reactive to minor deviations from the observed trend, and more volatile times in the observed data result in wider posterior intervals to account for increased uncertainty in the trend of the data.

Diagnostic plots, including MCMC trace plots, autocorrelation between the MCMC samples and pairwise correlation between parameters were examined and suggest the algorithm has converged. Further details on the posterior distributions of the model parameters, convergence and model diagnostics are provided in S3 Appendix.

**Table 1. Phase 1 versus Phase 2 median and 80% intervals for baseline parameters, $\mu_1$ and $\mu_2$.**

| Country | $\mu_1$ | $\mu_2$ |
|---|---|---|
| Italy | 4.39 (3.18,5.71) | 1.17 (0.69,1.8) |
| France | 4.57 (3.38,5.91) | 1.57 (0.97,2.28) |
| Spain | 5.78 (4.06,7.6) | 0.49 (0.28,0.76) |
| Germany | 4.17 (2.89,5.54) | 0.95 (0.59,1.39) |
| Sweden | 4.05 (2.88,5.44) | 1.79 (1.05,2.68) |
| U.K. | 4.51 (3.08,6) | 2.42 (1.32,3.75) |
| U.S. | 4.08 (3.13,5.15) | 4.1 (2.16,7.12) |
| China | 8.92 (6.29,11.73) | 0.82 (0.48,1.22) |
| Brazil | 4.18 (2.98,5.52) | - |
| India | 2.81 (2.02,3.72) | - |

**Table 2. Phase 1 versus Phase 2 median and 80% intervals for magnitude parameters, $\alpha_1$ and $\alpha_2$.**

| Country | $\alpha_1$ | $\alpha_2$ |
|---|---|---|
| Italy | 1.07 (1.05,1.09) | 0.94 (0.93,0.95) |
| France | 1.1 (1.08,1.11) | 0.92 (0.91,0.93) |
| Spain | 1.11 (1.09,1.13) | 0.96 (0.95,0.97) |
| Germany | 1.06 (1.03,1.09) | 0.91 (0.89,0.93) |
| Sweden | 1.07 (1.01,1.13) | 0.92 (0.89,0.95) |
| UK | 1.14 (1.11,1.17) | 0.95 (0.95,0.96) |
| US | 1.07 (1.06,1.07) | 0.97 (0.97,0.98) |
| China | 1.07 (1.01,1.15) | 0.8 (0.76,0.84) |
| Brazil | 1.03 (1.02,1.04) | - |
| India | 1.1 (1.07,1.13) | - |

Tables 1–3 present the posterior median and corresponding 80% posterior intervals for the model parameters. Further details for the other baseline parameter priors considered can be found in S4 Appendix. In most countries, the posterior interval for $\mu_2$ is consistently lower than $\mu_1$, indicating a reduction in the baseline rate of events from the beginning to later stages of the epidemic. The exception to this is the U.S. The results for the U.S. are highly sensitive to the prior choice; thus, wider priors return higher posterior estimates than expected when

**Table 3. Phase 1 versus Phase 2 median and 80% intervals for triggering kernel parameters, $\beta_1$ and $\beta_2$ and the means of their respective geometric distributions, $\beta_1^{-1}$ and $\beta_2^{-1}$.**

| Country | $\beta_1$ | $\beta_2$ | $\beta_1^{-1}$ | $\beta_2^{-1}$ |
|---|---|---|---|---|
| Italy | 0.88 (0.8,0.95) | 0.55 (0.48,0.63) | 1.136 (1.053,1.25) | 1.818 (1.587,2.083) |
| France | 0.97 (0.92,0.99) | 0.64 (0.58,0.7) | 1.031 (1.01,1.087) | 1.562 (1.429,1.724) |
| Spain | 0.96 (0.9,0.99) | 0.91 (0.85,0.95) | 1.042 (1.01,1.111) | 1.099 (1.053,1.176) |
| Germany | 0.65 (0.57,0.75) | 0.51 (0.45,0.59) | 1.538 (1.333,1.754) | 1.961 (1.695,2.222) |
| Sweden | 0.42 (0.32,0.54) | 0.5 (0.39,0.62) | 2.381 (1.852,3.125) | 2 (1.613,2.564) |
| UK | 0.79 (0.68,0.91) | 0.56 (0.5,0.62) | 1.266 (1.099,1.471) | 1.786 (1.613,2) |
| US | 0.99 (0.98,1) | 0.77 (0.66,0.89) | 1.01 (1,1.02) | 1.299 (1.124,1.515) |
| China | 0.4 (0.28,0.56) | 0.43 (0.35,0.54) | 2.5 (1.786,3.571) | 2.326 (1.852,2.857) |
| Brazil | 0.83 (0.73,0.93) | - | 1.205 (1.075,1.37) | - |
| India | 0.33 (0.26,0.41) | - | 3.03 (2.439,3.846) | - |

compared to other countries. In an earlier analysis, this behaviour was also prevalent for Sweden and the U.K., although it disappeared when considering a longer time series. This implies that there may be insufficient information in the data for the U.S. to reliably learn the model parameters for the second phase. However, without alternative data, it is not possible to improve modelling for the U.S. by considering a longer time series. This is due to a large anomaly at the end of the series, as discussed in the Data section. Nonetheless, it highlights the importance of having sufficient training data and being cautious when interpreting parameter estimates.

The magnitude parameter in the second phase, $\alpha_2$, is also consistently lower than the parameter for the first phase, $\alpha_1$. With a posterior probability (greater than 80%), it can be said for all countries that $\alpha_1 > 1$ and $\alpha_2 < 1$. This implies the process is explosive before the change point and becomes stationary after the change point, likely driven by the introduction of interventions to reduce the rate of infection.

The parameters for the geometric triggering kernel, $\beta_1$ and $\beta_2$, are similar for Sweden and China. However, for the remaining countries where two phases are considered, the kernel parameter for the first phase, $\beta_1$, is larger than $\beta_2$, indicating that the self-excitation has a longer memory in the second phase. For reference, $\beta = 0.4$ in the geometric kernel corresponds to an average of 2.5 days for the self-excitation, with the majority of the mass occurring within one week, whereas $\beta = 0.9$ is shorter, corresponding to an average self-excitation of just over 1 day with approximately 2 days of total memory.

## Model fit

Several measures are used to assess model fit. First, the model's capability to interpolate missing data is evaluated. Then in-sample and out-of-sample posterior predictive checks are considered. The purpose of prediction in this study is to assess model fit and to discover what can be learned about the process retrospectively.

The first measure of model fit considers how accurately the model can recover missing data. We randomly remove 10% of observations across the entire time series and treat the missing data as parameters in the model to estimate. Table 4 describes the number of interpolated data points for which the observed value lies within both the 95% and 80% credible intervals (CrI) of the posterior distributions for the missing data. Further details can be found in S5 and S6 Appendices. The proportion of data points correctly interpolated is generally high when considering the 95% credible intervals. This reduces when considering the 80% interval, however, is still high for most countries, capturing at least half of the missing data points.

**Table 4. Number of missing data points with actual value within 95% and 80% CrIs, out of the total number of missing data points.**

| Country | 95% CrI (average) | 80% CrI (average) |
|---|---|---|
| France | 11/14 | 7.4/14 |
| Italy | 13/15 | 11/15 |
| Germany | 13.4/14 | 10.2/14 |
| Spain | 8/11 | 6.2/11 |
| Sweden | 12.6/13 | 10.4/13 |
| U.K. | 11.8/14 | 9.2/14 |
| China | 8.6/9 | 7.2/9 |
| U.S. | 8.6/11 | 5.4/11 |
| Brazil | 6.6/8 | 4.6/8 |
| India | 7.8/9 | 6.8/9 |

The exception to this is the U.S., with just less than half of the missing data points accurately interpolated.

Prediction is a difficult task, particularly for complex phenomena such as the COVID-19 pandemic. For this particular model, more recent events have a larger impact on the intensity of the process. Thus prediction performed at a time where abnormal behaviour is occurring will be highly uncertain and often unreliable. Moreover, a prediction is only realistic in the short term and generally only at times where there is no evidence of abnormal behaviour. This is consistent with other models in the literature [37, 38, 62–64]. Thus we consider in-sample and out-of-sample posterior predictive checks in this study as a measure of model fit only.

In-sample prediction is performed by generating sample paths of the process for the range of model parameters obtained and comparing these to the observed time series. In particular, a random selection of posterior samples is taken, and the entire time series is simulated from these draws. The posterior predictive intervals from these simulations compared to the observed data are given in Fig 3. In general, the intervals for these simulations encapsulate or are very close to the observed data, however, they can be extremely wide and often underestimate the volume of events in the initial phase of the outbreak. This is likely due to variation in the assumed Poisson data generating distribution, and relatively wide priors on the baseline parameters for the first phase, resulting in a wide range of possible sample paths. Additionally, these sample paths did not adequately capture the observed trend in the U.S. However, we find that including the data from the first phase in the model and predicting the second phase results in improved accuracy of the posterior predictive intervals for all countries. These results are presented in Fig 4.

Out-of-sample (O.O.S.) validation is also performed for each country as a measure of model fit. First, we consider the initial phase of the epidemic before the change point. The model is trained on data from the first 15 days of the sample, followed by a 5-day O.O.S. prediction. We then repeat this process, increasing the length of the training period by 5 days until the change point. As shown in Fig 5, these predictions are reliable only in the short term, and become more unreliable as the end of the first phase approaches. The first phase predictions grow exponentially and quickly surpass the actual growth of the process, as the observed curve flattens due to the effects of preventative measures that have been implemented.

O.O.S. prediction is also considered for the second phase of the model, after the change point. We first train the model on data from the first phase and 15 days of the second phase. We then repeat the same procedure as described above with 10-day O.O.S. predictions. The downward trajectory of the infection cycle is more stable than the upward trajectory, so we consider a longer prediction duration. The posterior predictive intervals are generally very accurate for all countries, as seen in Fig 5. Compared to the O.O.S. validation performed for the first phase, the improvements in accuracy observed in the second phase are likely due to the stationarity of the process in the second phase, resulting in more predictable trends. For both phases, the accuracy of O.O.S. predictions depends on the endpoint of the training period for the model, and the type of behaviour preceding any predictions.

While we do not attempt to predict the course of the epidemic in this study, we do find that O.O.S. predictions may indicate when the peak in the number of events is approaching. This could be useful in countries that have not yet experienced a decline in the number of daily events, for example, Brazil and India in this study. Posterior predictive intervals that surpass the growth rate in the observed data indicate, and could pre-empt, the downward phase of the epidemic. Conversely, where the predictive intervals do encapsulate the observed data, it is unlikely that the peak is being approached. This is evident in Fig 5, where the curve for Brazil is flattening, resulting in unreliable O.O.S. predictions, compared to the more reliable predictions in India due to the strong upward trend.

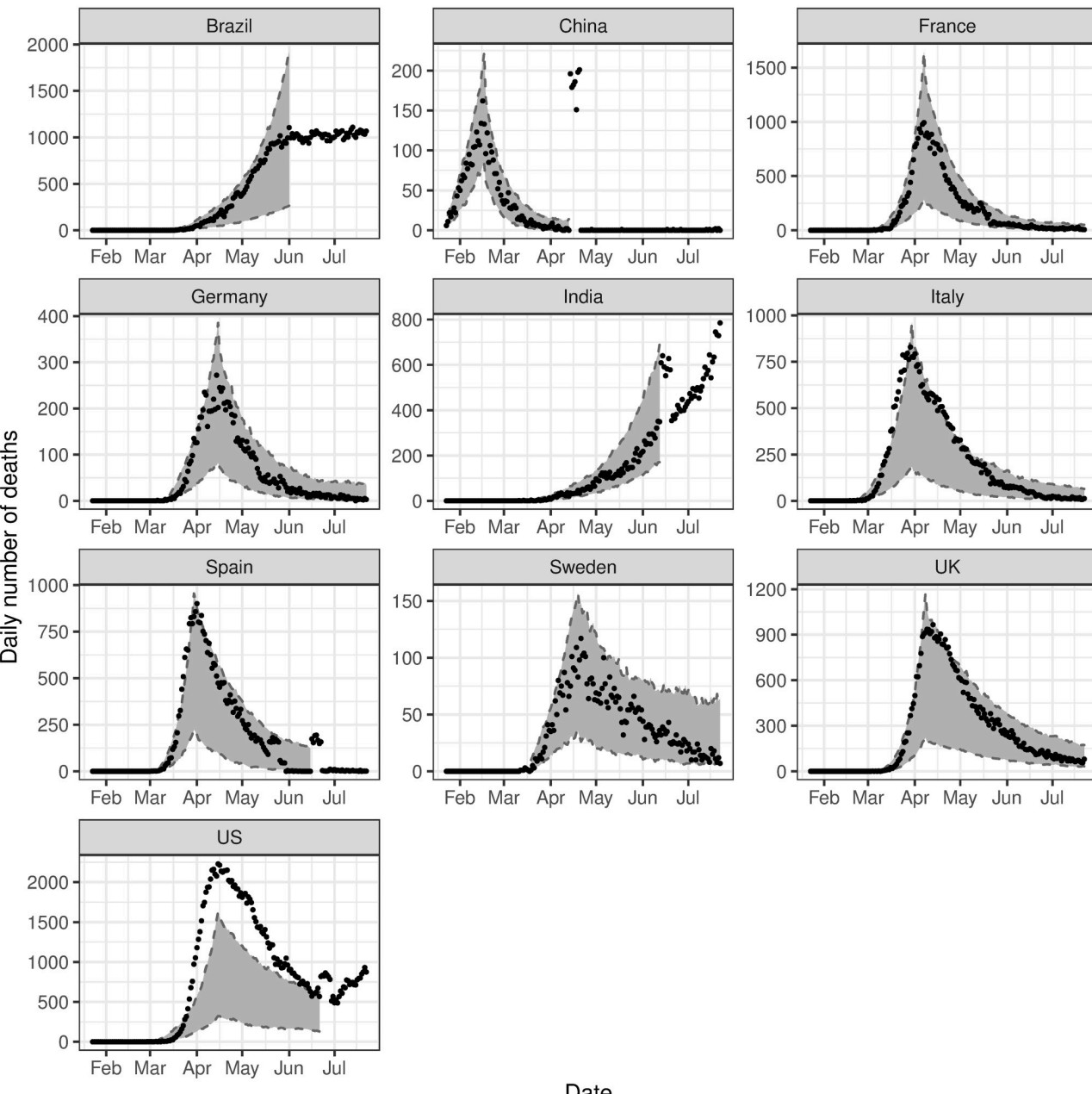

**Fig 3. In-sample validation.** The observed number of deaths (black dots) compared to the 95% posterior predictive interval for the estimated expected number of events, i.e. λ(t) (grey ribbon).

### Fitting subsequent phases

As the pandemic progresses further waves of infection, and thus deaths, are inevitable and will continue to be of interest for the foreseeable future, particularly as a vaccine is rolled out and new variants of the virus are discovered. There is no obvious endpoint to the pandemic, however it is of interest to investigate subsequent waves of infection as well. To address this, we extend our main analysis to determine whether our proposed model is applicable over a longer time period.

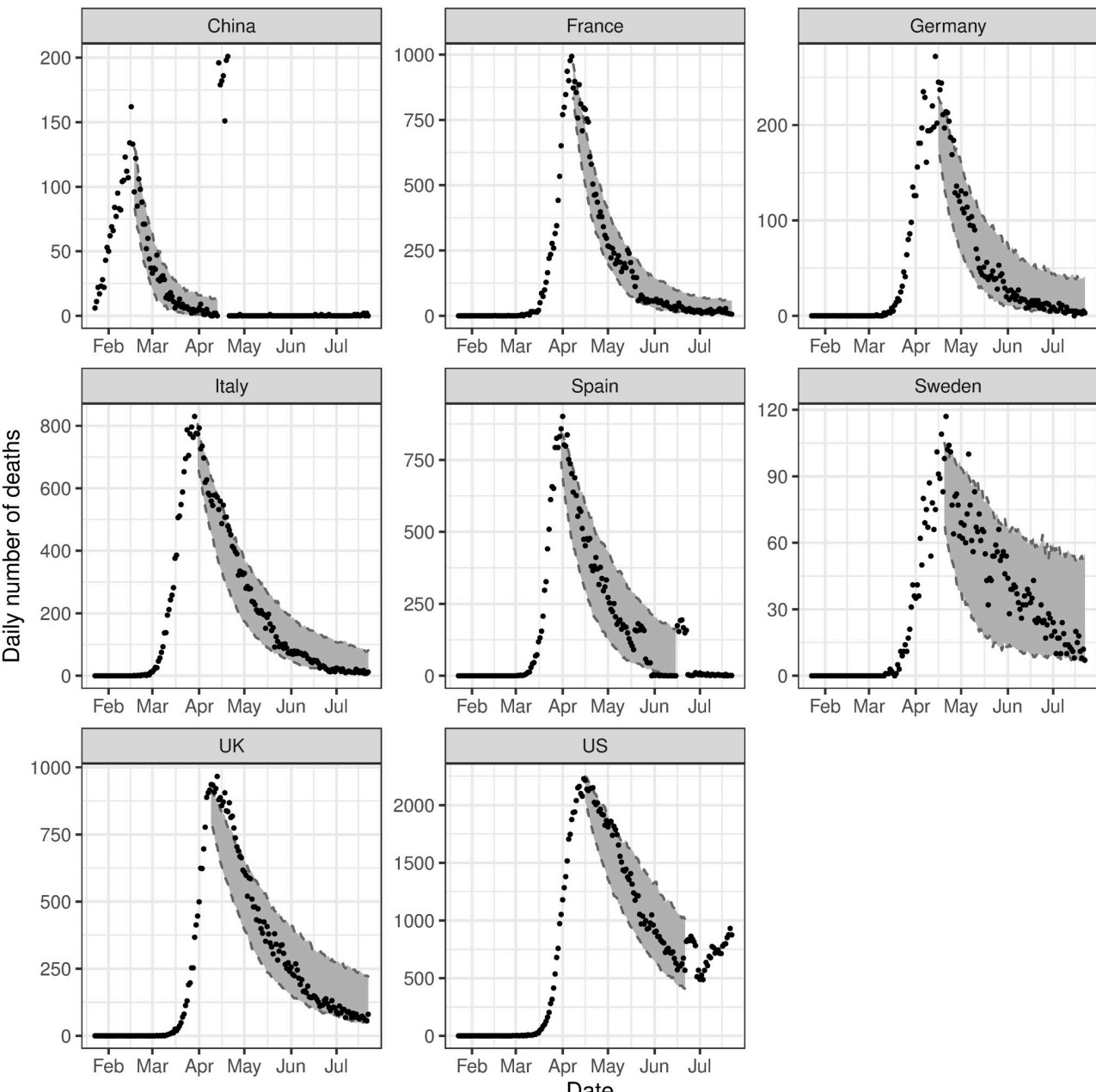

**Fig 4. In-sample validation, conditioned on data from the first phase.** The observed number of deaths (black dots) compared to the 95% posterior predictive interval for the estimated expected number of events, i.e. $\lambda(t)$ (grey ribbon).

We consider mortality data from the endpoint of our initial analysis, up to 4th February 2021. Countries with inadequate data to inform another phase were cut short. As such, the observation period for Brazil, U.K and U.S end on 7th January 2021, 24th January 2021 and 12th January 2021 respectively. Furthermore, for many countries there is a period of very low mortality in between the first and second waves of infection, and we do not consider this period. Additionally, China has not experienced a second wave, and thus it is excluded from this subsequent analysis.

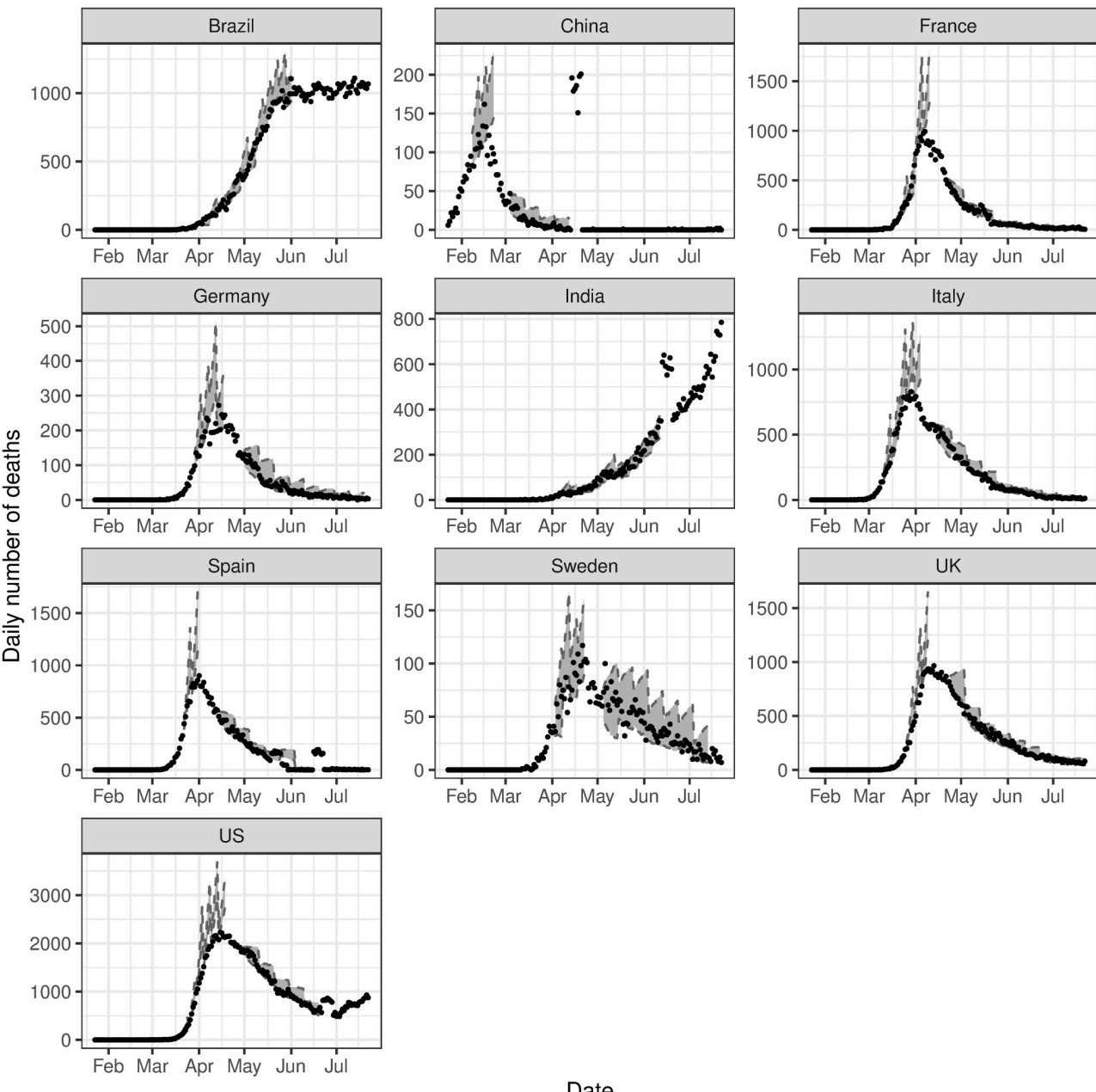

**Fig 5. Out-of-sample validation.** The observed number of deaths (black dots) compared to the 95% posterior predictive interval for the estimated expected number of events, i.e. $\lambda(t)$ using various training datasets (grey ribbons).

Change points were selected where there were obvious changes in the trajectory, in a similar fashion as the main analysis. The starting point of the second wave was selected as the time where either the 2 week or 4 week rolling average increases by 50% in a single week. The choice between a 2 or 4 week rolling average is chosen based on which more closely aligns to the start of the second wave upon visual inspection. We note that automatic change point detection algorithms such as the CUSUM algorithm [65] were considered, however, they are not appropriate for our model. These algorithms are generally based on the mean of the time series. Given the self-exciting nature of our model, changes in the intensity of the process do not

necessarily indicate changes in the underlying model parameters. The change points selected can be found in S7 Appendix.

Comparing the parameter estimates between the initial analysis and this subsequent analysis, several observations can be made. The full table of estimates can be found in S2 Table. Generally, while the baseline parameter $\mu$ in the initial analysis shows a reduction between the first and second phases, in subsequent phases the baseline mean begins to increase again. This is potentially due to the relaxing of restrictions and the opening of international borders. The magnitude parameter $\alpha$ acts as expected, in other words it is less than 1 for phases with a downward trajectory and greater than 1 for phases with an upward trajectory. In the initial analysis, $\beta$ is generally close to 1 in the first phase and reduces in subsequent phases.

Fig 6 shows the estimated intensity function against the observed data for the subsequent analysis. We find that the estimated intensity follows very closely to the observed data, as is also seen in the main analysis. We also consider in-sample (Fig 7) and out-of-sample validation (Fig 8), in the same manner as the main analysis. These both show promising results, with both in-sample and out-of-sample predictions aligning very closely to the observed data. The residuals, in this case referring to the difference between the observed data and the estimated intensity, for all phases in both the initial and subsequent analysis are provided in S8 Appendix, and show that the models for both sets of analyses are reasonable.

## Discussion

There are many strengths to our work, and some important considerations that needed to be made. We first discuss the main findings of this analysis. This is followed by detailing the limitations and potential extensions. Lastly we compare our model methodology to several popular approaches for modelling this type of phenomena.

### DTHP model

Infectious diseases have previously been studied using Hawkes processes. However, the scale, severity and uncertainty of the current COVID-19 pandemic make it a very challenging problem, providing a unique opportunity to evaluate the capacity of Hawkes processes in describing an incredibly complex process. Another source of complexity arises from the definition of what constitutes a COVID-19 death, which differs between countries. This analysis finds that by modifying the DTHP to incorporate change points, our model can adequately capture the overall process as distinct phases, while quickly reacting to and accommodating for some level of abnormal behaviour.

The findings of this work can also quantify the dynamics of these distinct phases in the pandemic. Our results from the initial analysis show that for the baseline parameters, the background rate in the second phase, $\mu_2$, is lower than that for the first phase, $\mu_1$. This is analogous to a reduction in the baseline level of exogenous events, possibly related to reduced travel and general mobility. Another factor could be increased levels of community transmission, affecting the self-exciting component of the intensity function, and thus placing less emphasis on the baseline component. In subsequent phases, $\mu$ begins to increase again, which suggests an increase in movement between countries. The exception to this is the U.S., for the reasons stated in previous sections. The baseline parameter could also be affected by the definition of a reported COVID-19 death, as this differs between countries. For example, when the criteria for reporting a death excludes cases where the person suffers from other illnesses in addition to the virus, this could result in an inflated baseline rate, as secondary events from unreported cases could be present in the data.

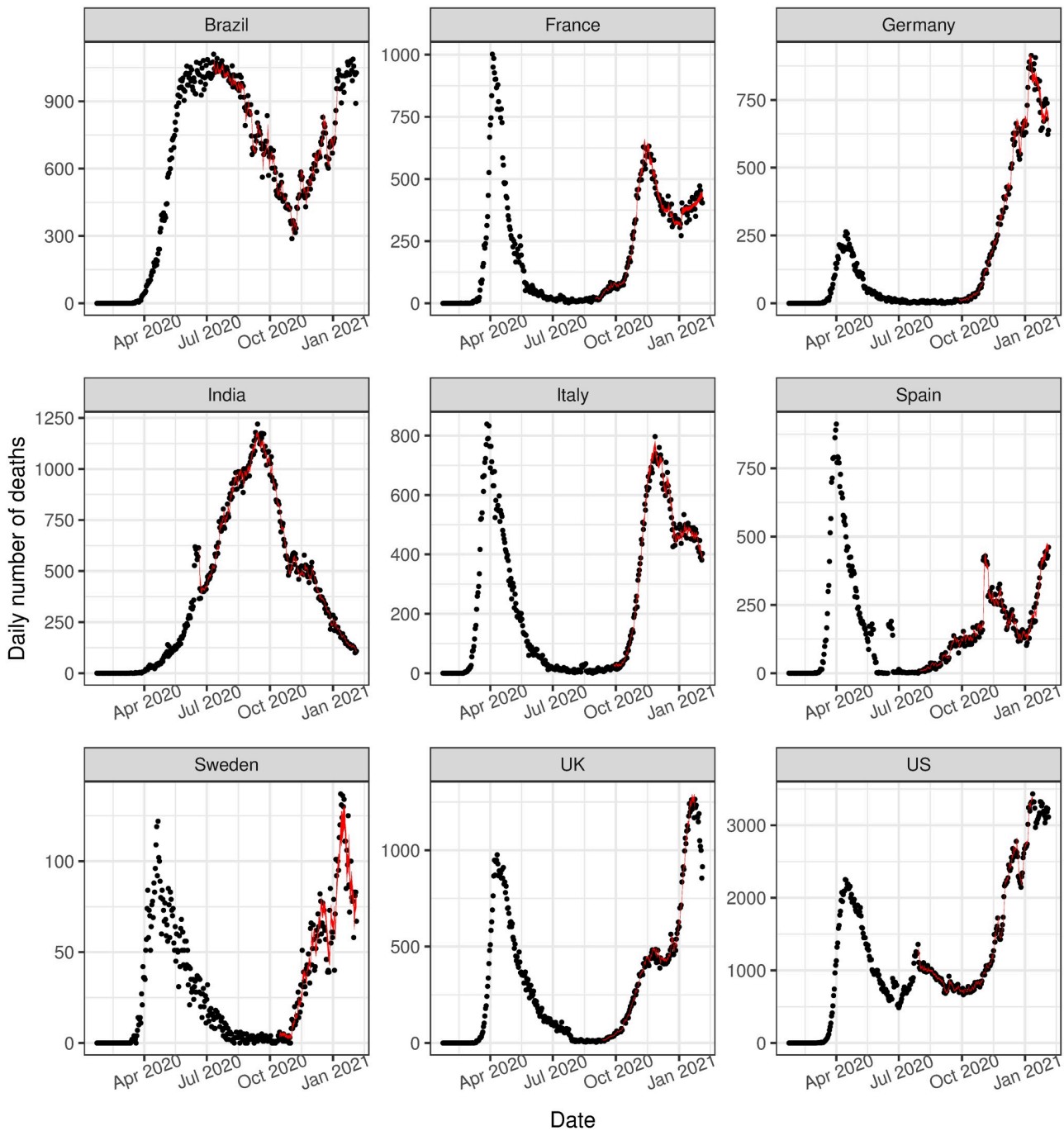

**Fig 6. Observed deaths versus estimated deaths (subsequent analysis).** The observed number of deaths (black dots) compared to the 95% posterior interval for the estimated expected number of events, i.e. $\lambda(t)$ (solid red ribbon), for the subsequent analysis.

Our initial results for the magnitude parameters show, with a high degree of certainty, that for the first phase $\alpha_1$ is greater than 1, and for the second phase $\alpha_2$ is less than 1. This exhibits the distinct differences between phases, as a magnitude parameter greater than 1 indicates the process itself is non-stationary, and similarly a magnitude parameter less than 1 suggests a stationary process. This pattern is also evident in the analysis of subsequent phases. We discuss

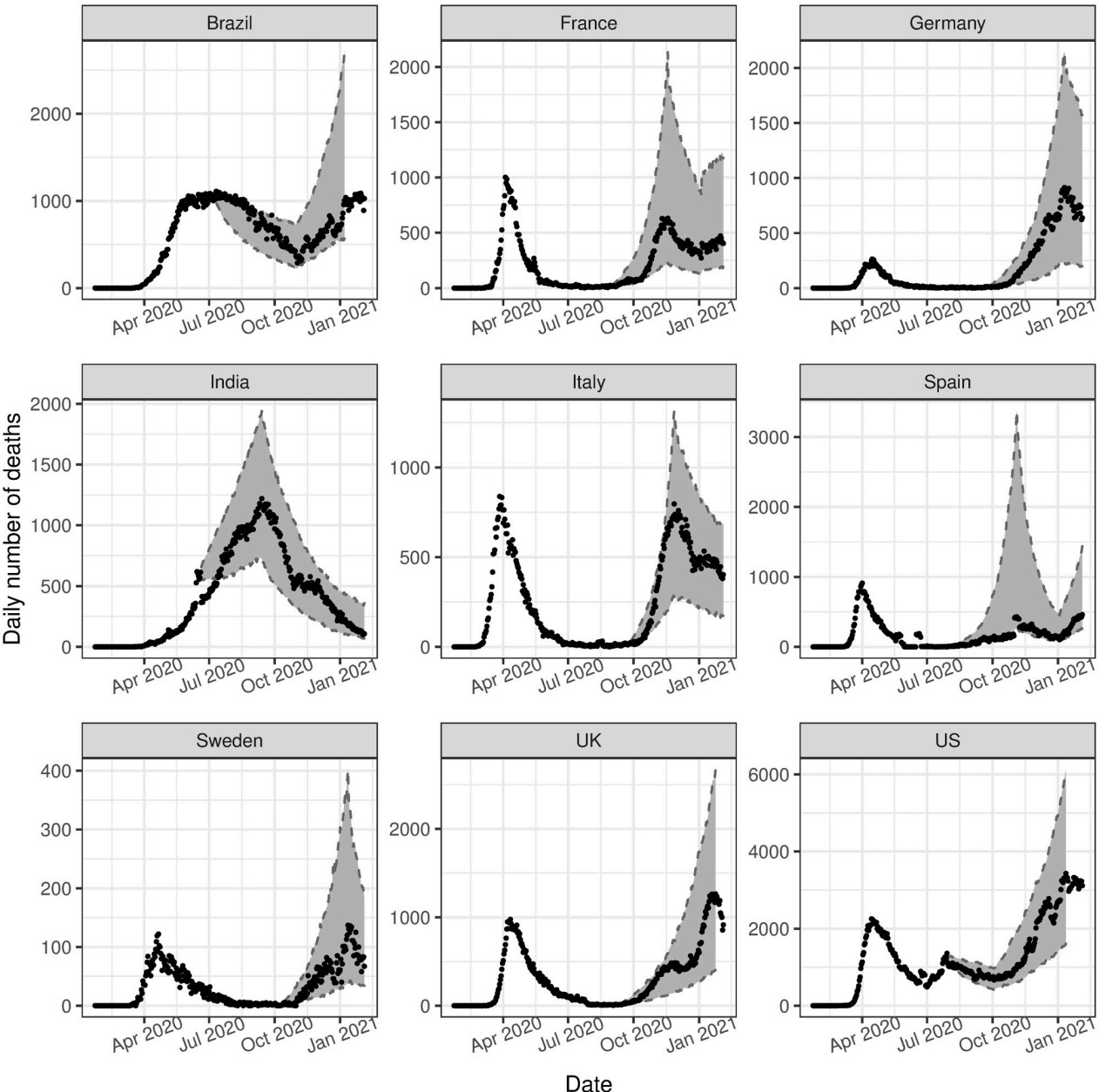

**Fig 7. In-sample validation for subsequent analysis, conditioned on data from the initial analysis.** The observed number of deaths (black dots) compared to the 95% posterior predictive interval for the estimated expected number of events, i.e. $\lambda(t)$ (grey ribbon), for the subsequent analysis.

below the similarities between the magnitude parameters in our model and the reproduction number in standard epidemiological models.

The triggering kernel parameter in the first phase, $\beta_1$, is higher than that for the second phase, namely $\beta_2$, for all countries except Sweden and China. This could suggest that in later stages of the epidemic when preventative measures have been implemented, the time between transmission is longer, as there is less opportunity for transmission. The two exceptions to this, Sweden and China, are on opposite ends of this spectrum. While China enforced very

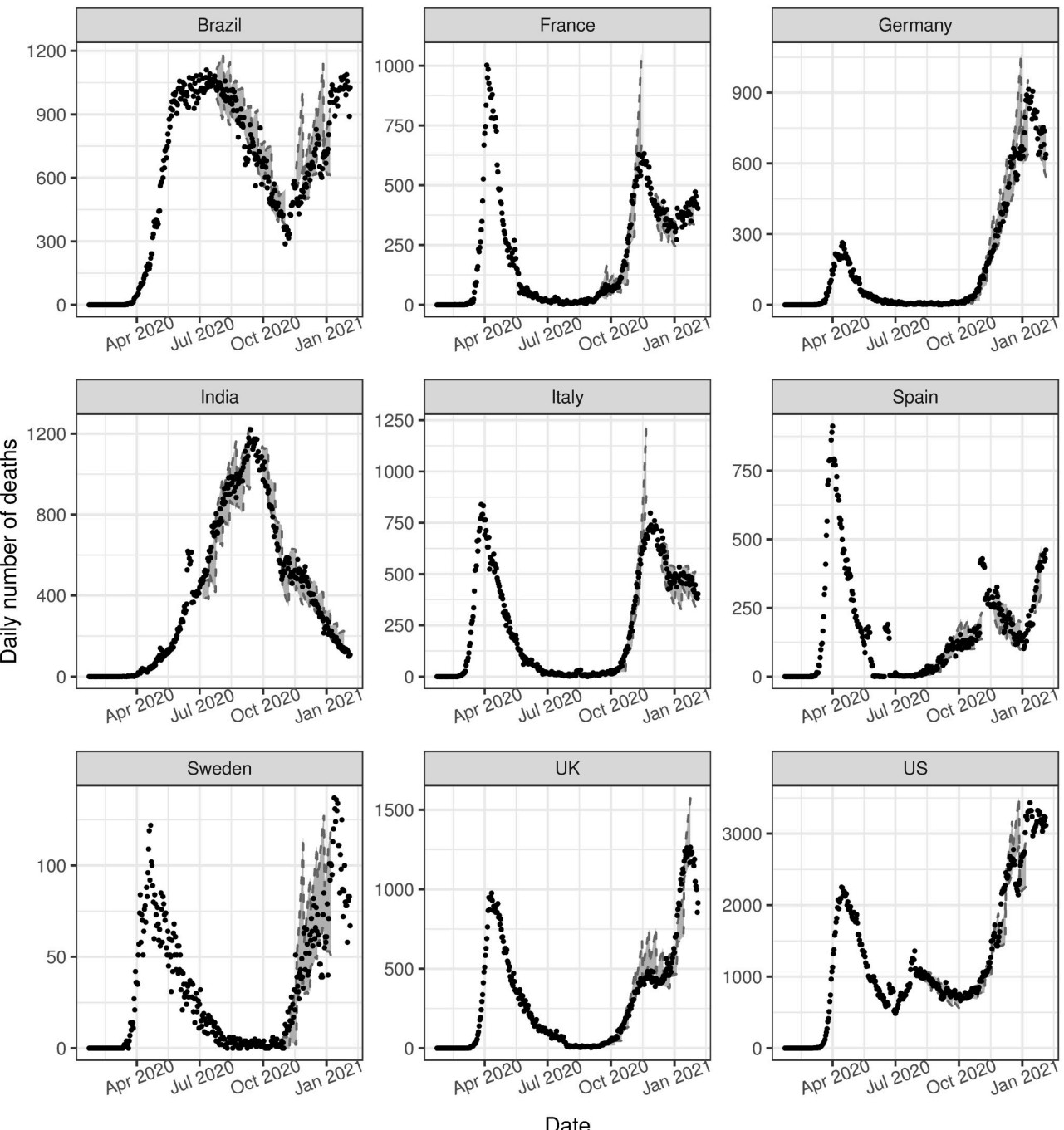

**Fig 8. Out-of-sample validation (subsequent analysis).** The observed number of deaths (black dots) compared to the 95% posterior predictive interval for the estimated expected number of events, i.e. $\lambda(t)$ using various training datasets (grey ribbons) for the subsequent analysis.

strict lockdown and quarantine requirements, Sweden adopted a soft approach to lockdown. Large $\beta_1$ values could also be an indication of instability in the initial phase of the pandemic, leading to difficulty in predicting and discerning patterns in the data. Additionally, this could be a result of death data being less reliable in early phases, as the process of counting COVID-19 deaths was not yet established.

Throughout the initial stage of this analysis, we have found difficulty in fitting the proposed model for the U.S. In particular, the posterior estimates for the baseline parameter are uncertain as they are heavily influenced by the prior choice. Additionally, in-sample posterior predictive checks found that the sample paths produced by the estimated model parameters do not resemble the observed trend. We consider the U.S. an anomaly, as their response to the virus by the relevant state-level authorities varied widely between states. While this is also true to an extent for other countries, the heterogeneity across the country was arguably more significant for the U.S., implying that the proposed model may need to be applied at a more granular level of regions to obtain more reliable results.

Despite our approach being able to accurately capture the dynamics of this complex process, we now address some limitations and extensions that could be considered. As the epidemic is still ongoing, new data is becoming available each day, and the model must be re-fit and tuned each time the data is updated. While we somewhat manually select change points in this analysis, an algorithm suitable to this model with automatic selection of the number of change points and their respective locations could also be considered. Additional change points need to be determined carefully as there must be sufficient information in each time series to inform parameter estimation. Another consideration is flexible Bayesian nonparametric splines [66] or other methods to provide time-varying parameters. However, the identifiability and existence of this model would need to be established. One could also consider different triggering kernels, including nonparametric kernels in order to improve the flexibility of the model. Another possible extension is considering covariates related to COVID-19 deaths, such as the number of people travelling and number of hospitals per capita.

## Comparison with other approaches

Here we discuss several of the many approaches that have been considered to model the ongoing COVID-19 epidemic, and the different perspectives they provide compared to our DTHP model. Compartmental models such as the SIR family of models are among the most popular methods for epidemic modelling. They are more detailed and consider the mechanics of the infection cycle, separating the population into categories such as susceptible, infected and recovered or deceased. Our DTHP model is simplified in the sense that we consider only death events. We chose to model deaths instead of infection numbers as the latter data was very unreliable in the beginning due to lack of testing and different testing policies across countries. However, as we show in S1 Appendix, as a first-order approximation, the death dynamics are helpful to understand the infection dynamics. This approximation is convenient when the infection data are unreliable, as occurred in the early stages of the COVID-19 pandemic. In the presence of data uncertainty such as this, the SIR model requires additional terms to account for this measurement error.

To compare the two frameworks, it is helpful to consider a stochastic variation of the SIR model as a bivariate Poisson process, comprised of infection and recovery events. Infection events are then governed by a Poisson process where the rate is based on the transmission rate and the current size of the susceptible and infected populations, corresponding to the rate of infection in the deterministic SIR model. Our model differs as we consider a discrete time scale, the daily number of events is Poisson-distributed and, conditioned on past events, the rate of events each day is given by Eq (2).

Another significant difference between our model and standard compartmental models is that the latter considers a finite population. In its original form, the Hawkes model assumes that there will be immigrant events arriving at a rate of the baseline mean $\mu$ indefinitely, implying an infinite population. However, finite population variants of the Hawkes model do exist

[43]. This differs from the SIR model, which naturally considers a finite population whereby the infection dies out once herd immunity is achieved. The impact of this difference is negligible in our modelling because we predominantly model the pandemic's initial phases, where not enough of the population has been infected or vaccinated to achieve herd immunity. This may not be the case for more prevalent diseases such as the flu, however both models are reasonable. As the flu season ends, there will still be new infections throughout the year, however on a smaller scale.

Hence our approach provides a simple model for unknown and volatile phenomena such as the COVID-19 pandemic, particularly in the early stages of the outbreak. Unlike the common flu, where the dynamics and course of infection are well understood and relatively predictable, COVID-19 is a new and unexplored domain. The various interventions that take place simultaneously result in complex interactions that complicate the dynamics of the process. Our focus is on the early stages of the epidemic where there is a great deal of uncertainty and volatility. The SIR model family is useful for phenomena where the mechanics are well known. However, complicated variants of these models are required to capture the complexity of this pandemic. Our simple model is useful in describing this early stage in the pandemic when there are still many unknowns. Our model also introduces randomness and flexibility that is not afforded in standard compartmental models. This allows our model to adapt to system changes induced by government interventions quickly.

The family of SIR models naturally follow the pattern of infections and deaths rising to a peak and then falling due to a reduction in the susceptible population. However, this is not the cause of the fall observed in the early stage of the pandemic. Instead, the fall is driven by external factors such as social distancing measures, temperature, and improvement in treatments, to name a few. SIR family models have also incorporated change points or time-varying parameters to account for these alternative drivers [51, 52]. Given our analysis's retrospective nature, the change points were quite obvious, and we did not estimate them. However, our Hawkes model can be easily augmented to induce this shape naturally. For example, we could consider a mixture of Hawkes processes for each of these distinct phases, estimate the unknown (or known) change points, or incorporate time-varying parameters.

Another more complex approach is that of agent-based modelling. These are more detailed than compartmental models, and are very useful if you have an understanding of the underlying mechanisms. Recent papers using this approach for the COVID-19 epidemic, referenced in the introduction, reveal the non-random nature of the underlying stochastic processes. Based on fluctuations in social participation and certain biological factors, they lead to the infection spreading, hospitalisation, and eventually to fluctuations of the fatality rate.

Alternatively, one could consider an even more straightforward approach, such as a piecewise exponential model. However, the Hawkes process allows for uncertainty in the model that is not possible with the exponential growth model, which is very strict and captures only the data trend. Allowing fluctuations in the data—particularly for volatile phenomena such as the current pandemic—is an essential aspect of providing a realistic model. The exponential model also becomes less appropriate as the pandemic progresses. In later phases, there are complex interactions that result in trajectories that are inherently not exponential. These are uncertain times, and our model strikes a balance between modelling the dynamics of the whole infection cycle and fitting a generic exponential model. We model some fluctuations motivated by the physical process, but with a simpler model than many others considered in the literature.

While there are many alternative approaches available, the Hawkes model is also a natural model for describing self-exciting phenomena. It provides a flexible and stochastic framework for modelling, and the parameters in our model provide interesting insights into the

pandemic. Namely, $\alpha$ is the average number of secondary infections and is related to the reproduction number, $\frac{\alpha}{\beta}$ is related to the average time an infected individual has infected someone, and $\mu$ relates to the occurrence of external excitations, or rather contaminations weighted by the probability of death given contamination. The $\beta$ parameter on its own also indicates how the time between infections changes throughout time.

The reproduction number, defined as the number of secondary infections from a single case, is a crucial parameter in epidemiological models. Similarly, the magnitude parameters in our model, given by $\alpha$, also represent the expected number of secondary cases caused by a single parent event. While their respective interpretations are similar at a superficial level, $\alpha$ is not directly comparable to reproduction numbers in epidemiological models. This is due to differences in model assumptions and the underlying mathematical frameworks, as our model's magnitude parameters do not provide the same information as the effective reproduction number. The effective reproduction number informs the level of herd immunity that will bring the virus under control, and the proportion of new infections that must be prevented to change the trend of events from increasing to decreasing [67], whereas our model parameters do not. However, we note that, similarly to reproduction numbers, if $\alpha > 1$ in our model there is exponential growth in the number of events and $\alpha < 1$ leads to a stationary model, which translates into a decrease in the number of deaths if the phase begins at a time with a high event intensity. We also consider a static variable that fundamentally averages over the whole period, rather than varying through time as the effective reproduction number would. We do this as reasonable change points were fairly obvious in the dataset used for this analysis. However, for more complex trajectories, other authors [44, 45] consider a Hawkes model with a time-varying magnitude parameter, which they refer to as a dimensionless reproduction number. This approach could inform the change point's location by observing when the magnitude parameter goes below 1. The change points could also be estimated, for example using the method suggested in [68].

Other key epidemiological parameters are generation times and serial intervals, which describe the time between infection and development of symptoms, respectively, for a pair of individuals. Our model does not capture this type of information, as we do not consider the relationship between specific pairs of individuals. As a result, it is not possible to obtain parameters such as growth rates, which are often of interest in epidemiological models. However, we can gain insight into an alternative temporal aspect of the contagion. The geometric triggering kernel in our model describes how the probability of contagion changes as time elapses. More precisely, we can determine, for a given day, the influence of past events on the expected number of events for that day.

## Conclusion

The utility of our model is not restricted to the current coronavirus epidemic, and could be used as a simple model to describe a much broader range of complex phenomena. We have demonstrated through this study that the proposed model is a simple, yet powerful tool for explaining an incredibly complex process. In general, models that attempt to describe complex processes can become increasingly complicated, as more intricate details are embedded and accounted for in the modelling. Thus having a parsimonious model that is flexible enough to competently capture the dynamics of a complex process, without adding too much additional complexity, is very desirable.

In particular for the current pandemic, this study shows that our simple discrete-time Hawkes process can capture the dynamics for different countries, despite the complexities involved with each country's unique response to the virus. The same underlying biological

process is affecting countries in different ways, and there is a significant difference in the impact and severity of the pandemic across different countries. Additionally, the actions that have been taken to stop the spread, and the timing of these also vary widely. These different behaviours between countries mean that the evolution of the pandemic for an individual country is very intricate within itself, and involves many unseen and complex hidden interactions that we cannot model directly. However, the proposed model, while being very simple, can capture these trends surprisingly well.

To adequately model the entire course of the pandemic, we find that we must make provisions as there are multiple distinct phases. Initially, there is exponential growth as the virus spreads, followed by a period of reduced infection rates as actions are taken to slow the spread. These distinct behavioural differences throughout the evolution of the epidemic must be acknowledged, as a single DTHP applied to the entire time series provides uninformative and uninterpretable parameter estimates. Hence a model that accounts for these different phases, such as the model presented in this work, is required.

Fitting a DTHP to the epidemic has led to some other unique insights. Our results show that a discrete-time model is appropriate for this application, avoiding unnecessary computational burden as well as additional noise due to artificial data imputation, as is required for the continuous-time model. This model also provides to an extent, interpretable parameters and an indication of the changing dynamics between distinct phases of the pandemic. We show that despite unique circumstances for individual countries, including the type and timing of non-pharmaceutical interventions, population demographics, and the overall impact of the virus, the model is flexible and can also accomodate some level of volatility in the data. Furthermore, one of the most surprising outcomes of this analysis is that, at the country level, a very simple DTHP model fits remarkably well to the number of deaths, thus capturing the dynamics of the COVID-19 pandemic.

## Supporting information

**S1 Appendix. Justification for Hawkes model on deaths.**
(PDF)

**S2 Appendix. About the average excitation duration.**
(PDF)

**S3 Appendix. Convergence and diagnostic plots for initial and subsequent analysis.** Top left hand panel: compares the observed number of deaths (black dots) with the 95% posterior interval for the estimated expected number of events (solid red ribbon). Top right hand panel: shows pairwise correlation between all parameters in the lower triangle, corresponding correlation values in the upper triangle, and the marginal posterior densities for each parameter on the diagonal. Bottom panel: shows trace plots on the top row and the autocorrelation function on the bottom row for each parameter. All figures were generated after thinning the posterior samples.
(PDF)

**S4 Appendix. Parameter estimates of baseline parameters for all prior choices.** Phase 1 versus Phase 2 median and 80% intervals of baseline parameters for countries with two phases.
(PDF)

**S5 Appendix. Missing data interpolation.** Tables containing number of missing data points with actual value within 80% and 95% posterior interval, for all prior choices.
(PDF)

**S6 Appendix. Figures from missing data interpolation.** The histogram represents the estimated posterior distributions for each of the missing data points. The black dashed lines show the 95% credible intervals around the posterior distributions. The solid blue line displays the observed number of deaths.
(PDF)

**S7 Appendix. Change point locations.**
(PDF)

**S8 Appendix. Plot of residuals.** For each country and phase, we calculate the estimated expected intensity of the process (i.e. $\lambda(t)$) using the samples of the parameter estimates obtained through the estimation procedure. The histograms then represent the median residual value (median of the difference between the observed number of events and the estimated expected intensity).
(PDF)

**S1 Table. Results from leave-future-out cross validation with Pareto smoothed importance sampling.** Expected log predictive density (ELPD) for a range of prior choices. Maximum ELPD in bold.
(PDF)

**S2 Table. Parameter estimates for original and subsequent analysis.** Comparison of median and 80% intervals of parameters for all phases, using the Gamma(5, 1) prior for $\mu$.
(PDF)

## Acknowledgments

The authors are grateful to Dr Gentry White, for helpful advice on modelling discrete-time Hawkes processes in the early stages of this project.

## Author Contributions

**Conceptualization:** Raiha Browning, Deborah Sulem, Kerrie Mengersen, Vincent Rivoirard, Judith Rousseau.

**Formal analysis:** Raiha Browning, Deborah Sulem.

**Methodology:** Raiha Browning, Deborah Sulem, Kerrie Mengersen, Vincent Rivoirard, Judith Rousseau.

**Supervision:** Kerrie Mengersen, Vincent Rivoirard, Judith Rousseau.

**Validation:** Raiha Browning.

**Visualization:** Raiha Browning.

**Writing – original draft:** Raiha Browning.

**Writing – review & editing:** Raiha Browning, Deborah Sulem, Kerrie Mengersen, Vincent Rivoirard, Judith Rousseau.

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
