## [Decision Letter · Decision Letter 0]

2 Jan 2021

PONE-D-20-34539

Simple discrete-time self-exciting models can describe complex dynamic processes: a case study of COVID-19

PLOS ONE

Dear Dr. Browning,

Thank you for submitting your manuscript to PLOS ONE. After careful consideration, we feel that it has merit but does not fully meet PLOS ONE’s publication criteria as it currently stands. Therefore, we invite you to submit a revised version of the manuscript that addresses the points raised during the review process.

We look forward to receiving your revised manuscript.

Kind regards,

Dan Braha

Academic Editor

PLOS ONE

Journal Requirements:

Reviewers' comments:

Reviewer's Responses to Questions

**Comments to the Author**

1. Is the manuscript technically sound, and do the data support the conclusions?

Reviewer #1: Yes

Reviewer #2: Yes

Reviewer #3: Yes

2. Has the statistical analysis been performed appropriately and rigorously? 

Reviewer #1: Yes

Reviewer #2: Yes

Reviewer #3: Yes

3. Have the authors made all data underlying the findings in their manuscript fully available?

Reviewer #1: Yes

Reviewer #2: Yes

Reviewer #3: Yes

4. Is the manuscript presented in an intelligible fashion and written in standard English?

Reviewer #1: Yes

Reviewer #2: Yes

Reviewer #3: Yes

5. Review Comments to the Author

Reviewer #1: In general, I have enjoyed reviewing this manuscript and think that it is worthy to be published in PLOS ONE. Though I want to raise few points, which I feel ought to be addressed prior to publication.

Comparison with SIR family models could be more detailed and focused on explaining how these different models are able to model the same phenomenon. One obvious difference, is that SIR considers finite population, while Hawkes process seems to not be aware of population being finite. In case of COVID-19, this might not be important, but to model other more prevalent diseases (e.g., flu), this could play a major role.

SIR family models seem to have advantage in that they have natural change point (when effective R becomes smaller than 1). Furthermore around this change point effective R seems to change continuously. In DTHP case, it seems that the change point is exogenously given prior to the fitting procedure. In practice, policy makers might not have this kind of knowledge.

I have recently seen another manuscript on Hawkes process [ https://arxiv.org/abs/2006.08355 ]. It explores different phases of the COVID-19 pandemic, which are not commonly taken into account by SIR family models: phase in which external excitation is present (cases incoming from abroad) and phase in which there is no external excitation. Your modeling seems to indicate that external excitation is negligible (besides first few cases)?

Definition of the excitation kernel uses different notation in definition (lines 190-191) and other equations in the manuscript. In most expressions the excitation kernel is function of time lag only, while in the definition it is a function of two values. Based on further text it is clear that "β" is a model parameter and that "i" represents time lag (which is confusing, because "i" is also used to index excitation events).

It is unclear what authors mean by "We choose the geometric kernel to resemble the exponential distribution". Why not exponential kernel then?

At the definition of the excitation kernel, it is not clear what parameter "β" means, from the later text it becomes clear that it controls average excitation time. Would it be possible to provide an equation for the average excitation time given this kernel?

Can DTHP be used for forecasting? How well it would perform?

Reviewer #2: In the context of statistical analysis of empirical data, this work belongs to a large corpus focusing on the applications of self-excitatory processes, here to COVID-19 epidemics. The authors used a modified discrete-time version with a memory to analyze the daily fatality rate, demonstrating it on the data from several countries with varying degree of social measures. In this approach, they are modelling the conditional mean (representing the fatality rate) without knowing about the underlying stochastic process (hence, considered as Poissonian).

Implementing a fixed breakpoint where the social measures take effect, they were able to detect, by determining the model parameters, how the dynamics of the initial growth phase differs from the decaying phase when the measures are effective. These findings make the primary value of this paper, which deserves the attention of the science community and may provide a deeper understanding of the importance of non-pharmaceutical measures to stop the epidemics.

The paper is very clearly written given this strictly statistical window. However, there are some aspects (mentioned below) that need to be discussed. Consequently, to broaden the view and help the reader to elucidate how these results can contribute. Specifically:

Recently, Agent-Based Modeling approaches of Covid19 epidemics revealed non-random nature of the underlying stochastic processes. Based on the fluctuations in social participation and certain biological factors, they lead to the infection spreading, hospitalization, and eventually to the fluctuations of the fatality rate [see Refs.: PLOS ONE 15(10) e0241163 (2020); Computers in Biology and Medicine (2020) 121, 103827; arXiv:2003.10218v1 (2020); Entropy 2020, 22(11), 1236;]

Given a generally considerable delay between the infection day of an individual and eventual fatal outcome, several factors can contribute (both individual and collective) along this timeline. Hence, the Poisson distribution in the context of fatality rate might be a reasonable approximation (still arguments need to be given). Then, the question remains if the same approach applied to the other two time series in the data (that is, the infection rate and recovery rate) would give an adequate description/with potentially different parameters?

Another question regards applicability of the analysis to the data from the second (and third) wave of the epidemics. In many countries, the developments beyond the data considered in this work are available. They reveal a different course of events, especially regarding the ICU and fatality in this epidemics. It is expected that the parameters and hence the epidemic's path could be different than in the first wave, the question is how different? Furthermore, if they could convey (in)efficiency of the social measures that we are currently experiencing, e.g., in Europe?

Reviewer #3: The authors present a model of the temporal dynamics of COVID-19 deaths based on a Hawkes process. The paper is very well written, and the analysis is sound. I would have no problem recommending the paper to be accepted as-is.

That being said, I do have one reservation that I recommend the authors address to improve accessibility of the paper and application of their proposed method in practice. This reservation stems from the basic question – why model COVID-19 deaths with a stochastic process when there is a low-dimensional dynamics apparent in the time-series data (e.g., Figure 1)? Here are my thoughts for the authors to consider:

• The number of infected individuals and deaths during an epidemic are known to proceed with the basic pattern shown in Figure 1, 2, etc. This shape has been well-modeled using the SIR model and its variants for many previous epidemics (e.g., Lipsitch et al. 2003). In the SIR model, this shape (a rise to a peak and then fall) results from the reduction of susceptible (S) individuals over time – that is, there are fewer deaths later in the epidemic, because there are fewer people left who have not yet been infected or have already died. Being a stochastic model, the Hawkes process doesn’t capture these longer time-scale underlying dynamics and the authors are forced to introduce a “change point” to effectively model the pre-peak and post-peak trajectories. While reading this paper, I was left wondering which is the appropriate modeling approach and does the Hawkes process model leave out important time-scales in the system? Can the authors comment more on how the Hawkes process model structure is linked to their presumptions about the underlying processes driving the death dynamics?

• Similarly, how does the Hawkes process model compare to simply a “piece-wise” exponential model (i.e., rate linear in x) with a change-point at the peak from a positive growth rate before the peak to a negative growth rate after the peak? If the peak is estimated from the data, this model would have 2 additional degrees of freedom. Are the small-scale fluctuations predicted by the Hawkes model worth the additional parameters?

Lipsitch, M., Cohen, T., Cooper, B., Robins, J. M., Ma, S., James, L., ... & Fisman, D. (2003). Transmission dynamics and control of severe acute respiratory syndrome. Science, 300(5627), 1966-1970.

6. PLOS authors have the option to publish the peer review history of their article (what does this mean?). If published, this will include your full peer review and any attached files.

Reviewer #1: No

Reviewer #2: **Yes: **Bosiljka Tadic

Reviewer #3: No

---

## [Author Response · Author response to Decision Letter 0]

23 Mar 2021

The authors thank the reviewers and editor for their time in considering our manuscript, and for their helpful comments and feedback. Please refer to the attached response document, where we have now addressed the comments in depth.

---

## [Editor Report · Decision Letter 1]

30 Mar 2021

Simple discrete-time self-exciting models can describe complex dynamic processes: a case study of COVID-19

PONE-D-20-34539R1

Dear Dr. Browning,

We’re pleased to inform you that your manuscript has been judged scientifically suitable for publication and will be formally accepted for publication once it meets all outstanding technical requirements.

Kind regards,

Prof. Dan Braha

Academic Editor

PLOS ONE

---

## [Editor Report · Acceptance letter]

1 Apr 2021

PONE-D-20-34539R1 

Simple discrete-time self-exciting models can describe complex dynamic processes: a case study of COVID-19  

Dear Dr. Browning:

I'm pleased to inform you that your manuscript has been deemed suitable for publication in PLOS ONE. Congratulations! Your manuscript is now with our production department. 

Kind regards, 

on behalf of

Professor Dan Braha 

Academic Editor

PLOS ONE